# Neutral Oil-Incorporated Liposomal Nanocarrier for Increased Skin Delivery of Ascorbic Acid

**DOI:** 10.3390/ma16062294

**Published:** 2023-03-13

**Authors:** Myoung Jin Ho, Hyun Jin Park, Myung Joo Kang

**Affiliations:** College of Pharmacy, Dankook University, 119 Dandae-ro, Dongnam-gu, Cheonan 330-714, Chungnam, Korea

**Keywords:** ascorbic acid, lipo-oil-some, edge activator, neutral oil, skin delivery, topical application

## Abstract

In this study, a neutral oil-incorporated liposomal system (lipo-oil-some, LOS) was designed to improve the skin absorption of ascorbic acid (Vit C), and the effects of an edge activator and neutral oil on the skin absorption of Vit C were evaluated. As components of the LOS system, sodium deoxycholate, polysorbate 80, and cholesterol were screened as edge activators, and camellia oil, tricaprylin, and grapeseed oil were employed as neutral oils. The LOS systems prepared by the ethanol injection method were spherical in shape, 130–350 nm in size, and had 4–27% Vit C loading efficiency (%). In a skin absorption study using a Franz diffusion cell mounted with porcine skin, the LOS system prepared with sodium deoxycholate (10 w/w% of phospholipid) exhibited 1.2-and 2.9-fold higher absorption than those prepared with polysorbate 80 and cholesterol, respectively. Moreover, the type of neutral oil had a marked effect on the absorption of Vit C; the liposomal system containing camellia oil provided 1.3 to 1.8 times higher flux (45.4 μg/cm^2^∙h) than vesicles with tricaprylin or grapeseed oil, respectively. The optimized lipid nanocarrier is expected to be a promising tool for promoting the skin absorption of Vit C and improving its dermatological functions.

## 1. Introduction

L-ascorbic acid (vitamin C, Vit C), a hydrophilic vitamin found in natural products, exerts various dermatological effects, such as improving skin conditions and anti-aging effects [1]. Vit C protects the skin from reactive oxygen species [2], promotes collagen biosynthesis, reduces melanin synthesis, and enhances skin immunity [3,4,5]. However, the low skin bioavailability of hydrophilic antioxidants limits their dermatological efficacy; only 8.1% of Vit C is absorbed through the skin after topical application [6]. The skin is composed of multiple layers, and the superficial stratum corneum (SC) acts as a key rate-limiting barrier for topically applied drugs. The SC layer comprises keratinized corneocytes and a tightly organized extracellular lipid matrix [7]. Tight junctions, located at the most apical part of the lateral cell-cell regions between epithelial cells, act as a physical barrier to the external environment. The tightly junctioned hydrophobic outer layer hinders the effective delivery of Vit C into the relevant skin layer [6].

Various cosmetic and pharmaceutical approaches have been investigated to overcome the limitations of skin absorption of Vit C, including the application of permeation enhancers and encapsulation into polymeric or lipid nanoparticles [8,9,10,11,12]. Liposomes, a well-established phospholipid-based vesicular system, are highly recommended for topical and transdermal delivery owing to their biocompatibility, degradability, and non-toxicity. The amphiphilic phospholipid bilayer creates a spherical vesicle that separates the aqueous core from the outer dispersant [13]. The amphiphilic characteristics of phospholipids enable the encapsulation of both hydrophilic and lipophilic drugs in the aqueous compartment and lipid bilayers, respectively [14]. When liposomal vesicles first contact the SC layer, the lipophilic membrane retains the vesicle on the SC layer surface and promotes “collision complex transfer.” Subsequently, the drug located between the liposomal bilayer or inner core is transferred to the SC layer, with the softening effect of the lipid compounds. Moreover, intact drug-loaded nanovesicles are able to penetrate the skin layer via inter-corneocyte pathways or appendageal pathways, such as hair follicles and sweat ducts [15].

Skin permeation of liposomal nanocarriers through the SC layer is affected considerably by the characteristics of lipid vesicles, including their particle size, surface charge, membrane fluidity, and lipophilicity. As the particle size of the liposomal vesicle decreases, penetration into the skin tends to increase for carboxyfluorescein-encapsulated liposomes [16]. Sen et al. reported that lipid vesicles with negatively charged phospholipids provided improved permeation through the porcine epidermis compared to that of neutral liposomal vesicles [17]. The membrane fluidity of liposomal vesicles can be increased by adding an edge activator to the lipid bilayer [18]. Single-chain surfactants, such as polysorbate 20 (Tween^®^ 20), polysorbate 80 (Tween^®^ 80, T80), sodium deoxycholate (Sod.DC), sodium cholate, Span 20, and Span 80 are employed as edge activators that decrease packing density and destabilize liposomal bilayers, thus increasing deformability. Elastic liposomes can adapt to mechanical stress and easily change their shape, facilitating the passage of elastic vesicles through the skin barrier [19]. In addition, the combination of absorption enhancers such as sulfoxides, laurocapram, terpenes, pyrrolidones, fatty acids, fatty alcohols, alcohols such as glycol, surfactants, and urea in the formulation can promote skin permeability [20]. Several pharmaceutical studies have reported the beneficial effects of natural essential oils and their components on skin absorption by decreasing the barrier effect of the SC layer through lipid perturbation via conformational permutations and phase separation [21].

In our previous study, we designed a novel liposomal carrier, named lipo-oil-some (LOS), containing neutral oil, phospholipids, and cholesterol (Chol) [22]. Incorporating lipophilic neutral oil (tricaprylin, Tric) between the liposomal bilayers markedly improved the photostability of the incorporated Vit C, with skin absorption comparable to that of the conventional liposomes. However, sophisticated formulation designs are required to optimize LOS systems for enhanced skin delivery. Therefore, we designed different LOS systems with several edge activators (Chol, Sod.DC, and T80) and neutral oils (Tric, camellia oil (CO), and grapeseed oil (GO)) to improve the skin absorption of Vit C (Figure 1). The physicochemical properties of the LOSs, namely morphology, particle size, surface charge, and Vit C loading efficiency and amount, were evaluated. Ex vivo skin absorption of Vit C-loaded LOSs with different edge activators and neutral oils was assessed using a vertical-type Franz diffusion cell mounted on porcine skin.

## 2. Materials and Methods

### 2.1. Materials

Ascorbic acid (Vit C), Sod.DC, Chol, Tric, GO, 1,2-dipalmitoyl-sn-glycero-3-phosphoglycerol (DPPG), sodium thiosulfate, formic acid, and succinic acid were obtained from Sigma-Aldrich (St. Louis, MO, USA). Phosphatidylcholine (PC) was acquired from Lipoid Korea (Geumcheon, Seoul, Korea). T80 and CO were purchased from Croda Korea (Seongnam, Gyeonggi-do, Korea) and Korea Similac (Pocheon, Gyeonggi-do, Korea). Ethanol (purity 99.5%) and acetonitrile (HPLC grade) were purchased from Duksan Reagents (Ansan, Gyeonggi-do, Korea) and J.T. Baker (Phillipsburg, NJ, USA), respectively. Other chemical reagents were of analytical grade and were used without additional refinement.

### 2.2. Fabrication of Vit C-Loaded LOS Systems

As previously reported, the Different Vit C-loaded LOS systems listed in Table 1 were fabricated using the ethanol injection method [23]. First, 200 mg of Vit C was dissolved in 10 mL of 10 mM succinate buffer (pH 3.0) via magnetic stirring to create an aqueous phase. Next, 200 mg of PC, 4 mg of DPPG, 20 mg of edge activators (Chol, Sod.DC, and T80), and 200–800 mg of oils (CO, GO, and Tric) were dissolved in 4 mL of ethanol solution to form the oil phase. The oil phase was then injected into the aqueous phase at 50 °C while stirring at 350 rpm for 10 min. The emulsified oil-in-water (o/w) mixtures were homogenized using a high-speed homogenizer at 15,000 rpm for 10 min to form nano-sized vesicles. The temperature was increased to 60 °C to evaporate the ethanol in the mixture, and the mixture was magnetically stirred for 30 min at 450 rpm. For the annealing process, the final LOS formulations were cooled to 20 °C for 120 min, transferred to sample vials, and kept at 4 °C. All procedures were performed under light-shielded conditions.

### 2.3. Physicochemical Characterization of LOS Formulations

#### 2.3.1. Morphology of Liposomal Formulations

The shapes of the LOS vesicles with different edge activators were examined using transmission electron microscopy (TEM; Talos L 120C; FEI, Hillsboro, OR, USA). Approximately 4 μL of each formulation was loaded onto a 200-mesh copper grid (Quantifoil Micro Tools GmbH, Jena, Germany), and manual blotting was performed at 4 °C for 1.5 s without extra staining process. Subsequently, the plunge-freezing method was employed for fixation by applying liquid ethane to the samples. The fixed samples were examined using TEM at an accelerated voltage of 120 kV [24].

#### 2.3.2. Liposomal Vesicle Size and Zeta Potential

The particle size and polydispersity index (PDI), an indicator of size reliability, of LOS systems were determined using a Zetasizer Nano (Malvern Instruments, Worcestershire, UK) and a dynamic light scattering particle size analyzer [25]. After a 20-fold dilution with distilled water, the samples were added to disposable cells (DTS0012, Malvern Instrument), and the vesicular size was measured at a 90° scattering angle. The zeta potentials of the LOS vesicles were also measured by loading 100 μL of 3-fold diluted samples onto a capillary zeta cell (DTS 1070; Malvern Instruments), and 20 repetitive runs were performed for each measurement. The measurements were performed in triplicate at 25 °C.

#### 2.3.3. Vit C Content Analysis

The Vit C content in the LOS formulation was measured using HPLC analysis. Triton X-100 (10%) was added and vortexed for 10 min to disrupt liposome bilayers and structure. The supernatant was then collected after centrifugation at 13,000 rpm for 10 min. The supernatant was diluted 2 times with formic acid (0.1%) and analyzed using a Waters HPLC system comprising an autosampler (Model 717 plus), pump (Model 515 pump), and UV–VIS detector (Model 486). The antioxidant was separated on a Kinetex EVO C18 analytical column (5 μm, 100 Å, 150 × 4.6 mm) under gradient elution conditions, with mobile phases of acetonitrile (A) and 0.1% formic acid (B) at a flow rate of 0.8 mL/min [26]. The initial setting was 0%(A), linearly increased to 80%(A) over 8.5 min, maintained isocratic for 0.5 min, and decreased to 0%(A) from 9–20 min. Vit C was eluted for approximately 3.5 min under this gradient condition. The column oven and autosampler temperatures were set at 25 °C and 4 °C, respectively. The injection volume was fixed to 30 μL, and the eluent was monitored at a wavelength of 245 nm. The calibration curve of Vit C was linear in the range of 1–100 μg/mL, with *r*^2^ of 0.999.

#### 2.3.4. Loading Efficiency and Amount of Vit C in LOS Systems

To determine the loading efficiency and amount of Vit C in the LOS systems, a stirred cell device (Amicon, Millipore, Billerica, MA, USA) and ultrafiltration discs (Biomax, Millipore, Billerica, MA, USA) with a 100 kDa pore size were applied. To obtain 60 mL for filtration, 3 mL of LOS samples was 20-fold diluted with 10 mM of succinate buffer (pH 3.0). The diluted solution was poured into the stirred cell, and nitrogen pressure was applied at a flow rate of 1 L/min. The filtrate containing the unloaded vit C molecules was then diluted 10 times with 0.1% formic acid and injected into the HPLC system to quantify the Vit C concentration. After calculating the amount of unincorporated Vit C, the Vit C loading efficiency was calculated using Equation (1) [27]. The loading amount was calculated by dividing the amount of loaded Vit C by the total liposomal composition (2).
(1)$$\mathrm{Loading}\;\mathrm{efficiency}\;(\%)=\frac{\mathrm{Initial}\;\mathrm{loaded}\;\mathrm{amount}\;\mathrm{of}\;\mathrm{Vit}\;C\;\left(\mathrm{mg}\right)\;-\;\mathrm{unloaded}\;\mathrm{amount}\;\mathrm{of}\;\mathrm{Vit}\;C(\mathrm{mg})}{\mathrm{Initial}\;\mathrm{loaded}\;\mathrm{amount}\;\mathrm{of}\;\mathrm{Vit}\;C\;(\mathrm{mg})}$$
(2)$$\mathrm{Loading}\;\mathrm{amount}\;(\mathrm{mg}/\mathrm{mg})=\frac{\mathrm{Initial}\;\mathrm{loaded}\;\mathrm{amount}\;\mathrm{of}\;\mathrm{Vit}\;C\;\left(\mathrm{mg}\right)\;-\;\mathrm{unloaded}\;\mathrm{amount}\;\mathrm{of}\;\mathrm{Vit}\;C\;(\mathrm{mg})}{\mathrm{Total}\;\mathrm{amount}\;\mathrm{of}\;\mathrm{Vit}\;C\;-\;\mathrm{loaded}\;\mathrm{LOS}\;\mathrm{composition}\;(\mathrm{mg})}$$

### 2.4. Ex Vivo Skin Absorption of Vit C-Loaded Lipo-Oil-Some (LOS) Formulations

A vertical-type Franz diffusion cell apparatus was used to assess the permeated and accumulated amounts of Vit C in LOS systems. As a skin model, 0.8–1.2 mm thickness of pig dorsal skin (Cronex Co., Ltd., Gyeonggi-do 16630, Korea) was adopted and fixed between the donor and receptor compartments. In the receptor compartment, 0.02% sodium thiosulfate was added to 10 mM succinate buffer (pH 3.0; stable pH for Vit C) to prevent skin disintegration. In the donor compartment, 0.2 mL of LOS formulation (4 mg as Vit C) was added, and the receptor phase was stirred at 450 rpm using a magnetic bar. Parafilm was applied at the top of the cell to prevent the evaporation of the loaded samples. The receptor solution was withdrawn at predetermined sampling points at 2, 4, 8, 12, and 24 h. After two-fold dilution with 0.1% formic acid, the Vit C concentration was analyzed using HPLC, as described above. To derive the permeation parameters, the permeated amount per unit area (μg/cm^2^) versus time (h) was plotted, and the flux (μg/cm^2^∙h), lag time (h), and permeability coefficient (10^−6^ cm/h) were obtained. The flux (μg/cm^2^∙h) was determined from the slope of the trend line of the permeated amount (μg/cm^2^) vs. time (h) after reaching a steady state. Trendline extrapolation of the permeated amount per unit area (μg/cm^2^) versus the time plot was used to calculate the lag time (h). The permeability coefficient (10^−6^ cm/h) was calculated by dividing the flux (μg/cm^2^∙h) by the concentration of Vit C in the donor compartment (μg/cm^3^) [28].

To measure the accumulated amount, the dorsal skin of the pig was recollected after 24 h, and both sides of the skin were wiped to remove the remaining samples and reagents. The skin was then diced into small fractions, added to 10 mL succinate buffer (10 mM, pH 3.0) containing sodium thiosulfate (0.02%), and vortexed for 24 h to extract the accumulated Vit C. To remove skin debris, the extract was centrifuged at 13,000 rpm for 10 min, and the supernatant was 2-fold diluted with formic acid (0.1%) for Vit C quantification using HPLC.

### 2.5. Statistical Analysis

Each experiment was performed at least three times, and the data are expressed as mean ± standard deviation. Significance was statistically analyzed using one-way analysis of variance (ANOVA) with a significance level of *p* < 0.05 unless otherwise indicated.

## 3. Results and Discussion

### 3.1. Formulation Strategy and Characterization of LOS Systems with Different Edge Activators

LOS systems with different edge activators, types, and amounts of neutral oils were fabricated using the ethanol injection method as previously described [22]. Ethanol injection is a simple and well-established process with high loading efficiency for hydrophilic compounds [29]. It is an easily scalable and simple manufacturing process with no harmful organic solvents, and hence, it is better than freeze-drying and film hydration methods [30]. In the ethanol injection method, PC, DPPG, different edge activators, and oily ingredients were dissolved in ethanol and injected into an aqueous medium containing Vit C. PC, a biocompatible, non-toxic phospholipid, is a primary component of the lipid bilayer, and negatively charged phosphatidylglycerol (DPPG) was included with PC to afford a negative charge to the liposomal surface, causing electrostatic repulsion between vesicles, thus preventing aggregation and precipitation of the colloidal system. Because Vit C is stable under weakly acidic conditions, the aqueous medium was buffered with succinate at pH 3–4. Homogeneous liposomal vesicles under 350 nm in size exhibiting enhanced skin absorption were fabricated using a homogenization process [16]. After homogenization, the remaining ethanol solvent was removed at 60 °C with stirring, and the dissolved lipid components solidified to form a liposomal structure. The labile antioxidant was stable during the fabrication procedure, maintaining over 95% drug content. The concentrations of PC, DPPG, edge activators, and neutral oils in the LOS system were 20, 0.4, 2, and 20–80 mg/mL, respectively.

First, LOS systems with different types of edge activators, Chol, Sod.DC, and T80 were prepared using CO as the neutral oil (L1, L2, and L3, Table 1). The amounts of Sod.DC and T80 (10% compared to PC) were sufficient to provide deformability of liposomal vesicles [31,32]. Liposomal vesicles containing 15% of the edge activator compared to phospholipids have been reported to provide more than 5 times higher deformability than rigid vehicles containing only 2% of the edge activator [31]. The morphological features of the different types of edge-activator-loaded LOS systems were examined using cryo-TEM (Figure 2). Liposomal vesicles were immobilized at −80 °C to prevent damage to the liposomal vesicles during pretreatment. In all formulations (L1, L2, and L3), spherical vesicles with a mixture of small unilamellar vesicles (SUVs), multilamellar vesicles (MLVs), and multivesicular vesicles (MVVs) were observed, regardless of the type of edge activator. Lipophilic neutral oil might act as an adhesive that connects individual membranes or vesicles to form multilayered or multivesicular structures. In L1, hydrophobic Chol was inserted between the phospholipid bilayers to increase the rigidity of the vesicular membrane. In contrast, for L2 and L3, the hydrophobic steroid nucleus moiety and sorbitan monooleate group were imported between the bilayers. The insertion of Sod.DC and T80 into the lipid bilayer may alter the fatty acid chain packing and enhance membrane fluidity. Interestingly, in the L2 system, oil-in-water emulsion droplets were also observed in liposomal vesicles. As reported in a previous study, the high compatibility of CO with T80 promotes the formation of emulsion droplets, along with its incorporation into the liposomal bilayer [33]. Although the shapes of the vesicles varied (SUVs, MLVs, and MVVs), the vesicular sizes in L1, L2, and L3 were uniform between 100 and 200 nm. The vesicular size determined using the dynamic light scattering method (Zetasizer Nano) coincided with cryo-TEM observations, exhibiting an average hydrodynamic diameter of 127–197 nm and homogeneous distribution (Figure 3A). The shearing force during the homogenization process provided homogeneous nanosized vesicles with PDI values below 0.46. The surface charges of L1, L2, and L3 vesicles were determined to be −63, −75, and −65 mV, respectively. The LOS system containing Sod.DC (L2) had a 10–12 mV lower zeta potential than those containing Chol and T80, possibly because of the anionic carboxyl group of Sod.DC. This result is in agreement with a previous report that Sod.DC-containing ultra-deformable vesicles exhibited an 11 mV lower zeta potential than T80-containing vesicles [34].

The loading efficiency and amount of Vit C in the different edge activator-incorporated LOS systems are shown in Figure 3B. The loading efficiencies of L1, L2, and L3 were 27, 18, and 12%, respectively. Accordingly, the loading amounts of L1, L2, and L3 were determined as 0.08, 0.06, and 0.03 mg Vit C/mg LOS, respectively. Chol (L1) may increase the rigidity of the liposomal membrane, preventing leakage of Vit C to external media during the manufacturing process. In the case of L2 containing Sod.DC, despite the increased fluidity of the membrane, the loading efficiency was over 18%. The hydrophobic neutral oil (CO) incorporated between the bilayers prevents the leakage of Vit C. In contrast, in the case of the LOS containing T80 (L3), the loading efficiency was 12% because of the formation of oil droplets that could not load the hydrophilic substance, as observed using cryo-TEM (Figure 2). Moreover, the polyoxylated group on the liposomal surface causes steric hindrance, hindering drug loading in the aqueous compartment.

### 3.2. Ex Vivo Skin Absorption Profiles of LOS Systems with Different Edge Activators

Skin permeation and accumulation profiles of LOS systems with different edge activators were evaluated using a Franz diffusion cell. Porcine skin is widely used for ex vivo transdermal delivery studies owing to its morphological and anatomical similarities to human skin. Porcine skin has an SC thickness of 21–26 μm; its lamellar organization, hair follicle density, and permeability profile for several chemical compounds are comparable to that of human skin [35,36]. Furthermore, its low individual deviation and easy availability mean that porcine skin is the best ex vivo tissue alternative to human skin [37].

The skin permeation profiles and relevant permeability parameters of the LOS systems (L1, L2, and L3) are shown in Figure 4A and Table 2, respectively. Following topical application, Vit C molecules began to permeate the skin rapidly (lag time of 0.4 and 1.32 h), and the amount of Vit C determined in the receptor phase increased linearly during the experimental period in all LOSs with different edge activators. L2 or L3 systems containing edge activators such as Sod.DC or T80 exhibited higher permeability than L1 (Chol), and the amount of Vit C permeated 24 h after application of L1, L2, and L3 was determined to be 319, 1080, and 845 μg/cm^2^, respectively (Figure 4A). The flux value of L1, L2, and L3 after reaching steady-state were calculated to be 13.0, 45.4, and 38.7 μg/cm^2^∙h, respectively. The permeability coefficient (10^−6^∙cm/h) values of L2 and L3 were also 3.5 and 3.0 times higher than those of L1 (Chol), respectively (Table 2). The percentages of Vit C permeated or deposited in the skin compared with the applied dose are depicted in Figure 4B. The amount of Vit C accumulated after 24 h ranged from 3.4–4.0 percent of the applied dose, showing no differences among L1, L2, and L3. However, as shown in Figure 4A, there was a marked difference in permeation, and the amount of Vit C permeated with L2 and L3 was 47% and 37%, respectively, which was 3.4 and 2.6 times higher than that of L1, respectively. The sum of the permeation and deposition for L2 and L3 was calculated to be 50 and 41%, respectively, which was significantly higher than that of L1 (17%). L2 and L3 prepared with Sod.DC or T80 possess the characteristic deformability to squeeze across the micro-lamellar spaces among keratinocytes and more effectively transfer the loaded Vit C to the inner skin layer. Moreover, the presence of edge activators with high HLB (hydrophilic–lipophilic balance)s attracts water around vesicles when applied to the skin. Transdermal hydration gradients, caused by the evaporation of surface water after topical application, act as a driving force for vesicle permeation [18]. During this process, the deformable L2 and L3 systems with improved membrane fluidity can easily change their shape and permeate the skin with less resistance than a rigid structure (L1). The HLB values of Sod.DC and T80 were estimated to be 16 and 15, respectively, whereas cholesterol had no HLB value owing to its high lipophilicity, with a log *p* value of 8.7 [38,39]. In particular, the permeated amount and flux value of the Sod.DC-incorporated formulation (L2) were 1.3 times and 2.3 times higher than those of the T80 (L3)-incorporated LOS systems, despite no statistical significance (Table 2). This agrees with a previous study showing that the repulsive effect of negatively charged liposomes creates a wide gap between corneocytes and increases transdermal efficiency [40]. Owing to their polar surface charge, they are expected to attract large amounts of water, thereby increasing the movement of vesicles along osmotic gradients [41]. In line with these results, several studies have enhanced the permeation profiles of carvedilol intranasally, tamsulosin, and asiatic acid-containing transferosomes using Sod.DC as an edge activator [32,42,43]. Therefore, we selected Sod.DC as the edge activator and fabricated additional LOS formulations for optimization.

### 3.3. Characterization of LOS Systems with Different Neutral Oils

After selecting Sod.DC as the edge activator, different LOS systems with neutral oils, including CO, GO, and Tric, at concentrations of 2% *w/v* were constructed. When LOSs were prepared with CO or GO at concentrations over 30 mg/mL, the neutral oils were not completely incorporated into the lipid bilayer, and the excess oils floated on the upper layer. Therefore, LOS systems containing neutral oil at a concentration of 20 mg/mL were designed, and their physicochemical properties were evaluated.

The vesicular size and PDI of LOS with different neutral oils are shown in Figure 3C. The vesicles of the Tric-incorporated LOS systems were relatively larger (293 nm) compared to those of the CO- and GO-containing LOS systems (190–220 nm). Tric, a medium-chain triglyceride, has adhesive properties that tighten the adhesion of each vesicle [44,45], forming multilayered and/or multivesicular liposomes. The PDI values of L2, L4, and L5 ranged between 0.2 and 0.3, indicating that they were fabricated uniformly. 

With respect to the Vit C loading efficiency and amount, CO-incorporated L2 showed a substantially larger loading efficiency of 18.5%, which was 4.6 and 2.1 times larger than that of L4 and L5, respectively (Figure 3D). This suggests that the morphological and structural characteristics of the fatty acids contained in neutral oils could affect the loading or leakage of incorporated Vit C. CO mostly consists of oleic acid (C18:1) (83.6%), a monounsaturated fatty acid (MUFA), which has only one unsaturated carbon bond among the long fatty chains [46]. Oleic acid from CO is expected to be located inside the lipid bilayer of the vesicular membrane in a well-aligned form, as reported previously [47,48]. However, a linoleic acid (C18:2), a polyunsaturated fatty acid (PUFA) that contains two double bonds at the sixth and ninth carbon, is abundantly included in the GO (74.7%) [49]. It has been reported to create a more curved structure than MUFA in lipid bilayers [50]. Therefore, GO- incorporating L4 is expected to have a sparser membrane structure with a low loading efficiency. In contrast, the Tric-incorporated L5 exhibited a low loading efficiency of 8.8% (Figure 3D). Tric molecules with three-pronged structures and high lipophilicity might form loose or fenestrated membranes. The zeta potentials of L2, L4, and L5 were similar (−75, −78, and −73 mV), as the oily ingredients incorporated into LOS systems were neutral under acidic conditions (pH 3).

### 3.4. Ex Vivo Skin Absorption Profiles of LOS Systems with Different Neutral Oils

Skin permeation and accumulation of Vit C following topical application of LOSs with different neutral oils were further evaluated using an ex vivo porcine skin model. The permeation profiles and parameters are shown in Figure 5A and Table 3, respectively. The total percentage of Vit C permeated or deposited 24 h post-administration is depicted in Figure 5B. CO and GO have been exploited as new oil candidates for LOS systems owing to their various cosmeceutical and transdermal applications [51,52].

All LOS formulations containing different oily ingredients (L4, L5, and L6) exhibited rapid skin permeation, and the amount of Vit C permeated increased linearly for 24 h. L2 (CO), L4 (GO), and L5 (Tric) provided higher permeability of Vit C than Chol-containing rigid liposomes (L1), providing approximately 3.5, 1.9, and 2.7 times higher flux values, respectively. Among them, CO-incorporated LOS (L2) provided the highest permeation of Vit C; the permeated amount of L2 was approximately 1.8 and 1.2 times higher than those of GO (L4)- and Tric (L5)-incorporated LOS systems, respectively. CO is a well-known permeation enhancer in cosmetics and pharmaceutical transdermal delivery [53]. When skins were treated with CO before the transdermal application of flurbiprofen and diclofenac sodium gel, the permeation profile remarkably increased compared with that of untreated skins [50]. Oleic acid, the main component of CO, is reported to not only initiate a selective perturbation of the intracellular lipid bilayers present in the SC but also increase the deformability of liposomal membranes [54,55]. Therefore, when the CO-incorporated LOS formulation (L2) was applied to the skin, both the liposomal vesicle itself and unencapsulated Vit C permeated into the SC tissues with the help of CO’s synergistic effect, showing a total accumulated and permeated percentage of 50.3%, which is 1.8 and 1.3 times larger than those of L4 and L5, respectively. Considering the permeability profiles and physicochemical properties, CO was selected as the optimal oily compound for LOS systems.

### 3.5. Characterization of LOS Systems with Different Amounts of Tric

The effect of the amount of neutral oil on the absorption of LOS by the skin was evaluated using Tric in the 20–80 mg/mL range. In the case of CO, despite the excellent permeability of Vit C, a LOS system with a higher CO content of over 3% was not fabricated homogeneously due to the floating of excess oil. Conversely, in the case of Tric, 5 and 8% oil-containing LOS systems (L6 and L7, respectively) were fabricated with no observation of floating of the unincorporated excessive oils. Therefore, LOS containing various concentrations of Tric was prepared, and the effect on skin absorption was evaluated.

The vesicular size and PDI of LOSs prepared with different amounts of tris are depicted in Figure 3E. The vesicular size tended to increase as the amount of Tric increased; the vesicular sizes of L2, L6, and L7 were determined to be 293, 305, and 349 nm, respectively. As the amount of Tric increased from 20 mg/mL to 80 mg/mL, more Tric molecules might be proportionally inserted into the lipid bilayer by pulling the adjacent PC molecules laterally, increasing the vesicular size. The amount of Tric did not significantly affect the loading efficiency of Vit C; the loading efficiency of Vit C in L2, L6, and L7 ranged from 8.8 to 9.6%. In addition, with respect to the surface charge, the effect of the amount of Tric, a neutral oil, was negligible, with a similar zeta potential between −80 and −70 mV.

### 3.6. Ex Vivo Skin Absorption Profiles of LOS Systems with Different Amounts of Tric

The effect of the amount of Tric on Vit C absorption by the LOS system was evaluated at concentrations of 2% (L5), 5% (L6), and 8% (L7). The permeation profiles and parameters are shown in Figure 6A and Table 4, respectively. The total percentage of Vit C permeated or deposited 24 h post-administration is depicted in Figure 6B. The increase in Tric content in the LOS system did not contribute to the increase in skin permeation of Vit C; the flux values of Tric 2%, 5%, and 8% formulas were determined to be 35.5, 22.6, and 23.9 μg/cm^2^∙h, respectively. Although there was no statistically significant difference, the Tric 2% LOS (L5) formulation provided 1.6 and 2.1-fold higher flux values than the Tric 5% (L6) and 8% (L7) formulations, respectively. The permeability coefficient of the Tric 2% LOS (L5) was also 1.6 and 1.5-fold higher than those of the 5% and 8% formulations, respectively. Accordingly, the total amount of Vit C absorbed was also higher in L5 (40%) than in L6 (25%) and L7 (27%). Tric, a type of medium-chain triglyceride, is a lipophilic compound but skin-absorbable oil which has been reported to have skin-softening properties. Accordingly, the Tric contained in the LOS might act as a permeation facilitator through its lipid-softening property or by promoting transfer between the drug in the liposomal bilayer and the stratum corneum. However, when excess amounts of neutral oil are applied to the skin, the excess oil may form a thin film on the skin's surface, resulting in low skin permeation of the hydrophilic Vit C molecules. Actually, the lag time of L7 prepared with 8% Tric was delayed to 3.1 h. From these findings, we concluded that the appropriate content of Tric in the LOS system is 20 mg/mL for effective skin delivery of Vit C.

## 4. Conclusions

In this study, a refined LOS system was constructed by varying the edge activators and neutral oils for the effective skin delivery of Vit C. The LOS system containing Sod.DC (20 mg/mL) and CO (20 mg/mL) as the edge activator and neutral oil, respectively, provided a high flux value of over 45.4 ug/cm^2^∙h and delivered 50% of the loaded dose to the skin in an ex vivo skin absorption study. The optimized LOS vesicles were successfully characterized as spherical in shape, 196 nm in size, and having -75 mV of a surface charge. We expect the novel liposomal system to be an effective tool for the skin delivery of Vit C with an improved absorption-promoting effect.

## Figures and Tables

**Figure 1 materials-16-02294-f001:**
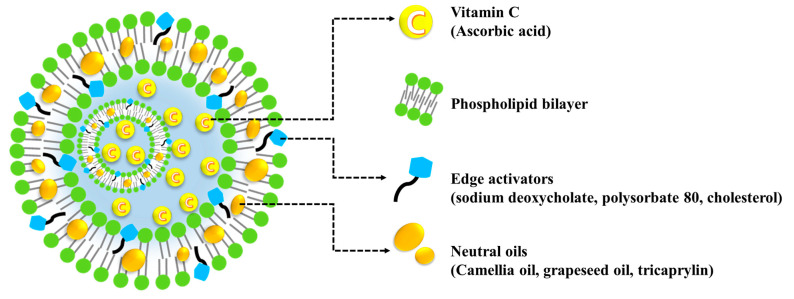
Schematic illustration of the configuration of Vit C-loaded lipo-oil-somal nanocarrier.

**Figure 2 materials-16-02294-f002:**
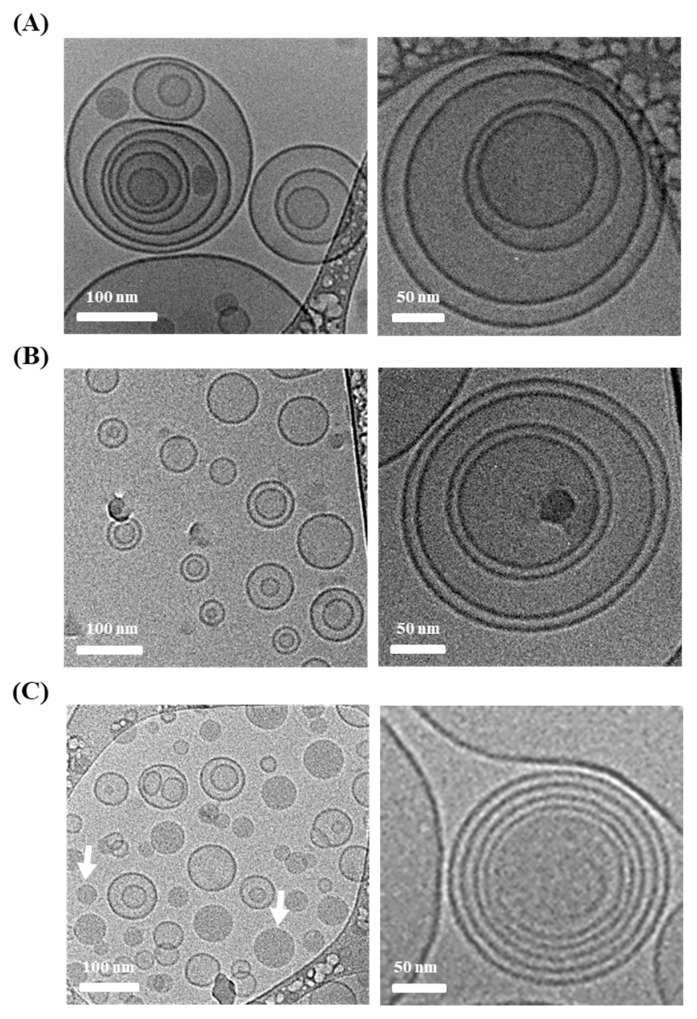
Representative cryo-TEM images of lipo-oil-somes (LOSs) prepared with different edge activators; (**A**) cholesterol (L1), (**B**) sodium deoxycholate (L2), and (**C**) polysorbate 80 (L3). White arrows in (**C**) indicate oil-in-water emulsion droplets.

**Figure 3 materials-16-02294-f003:**
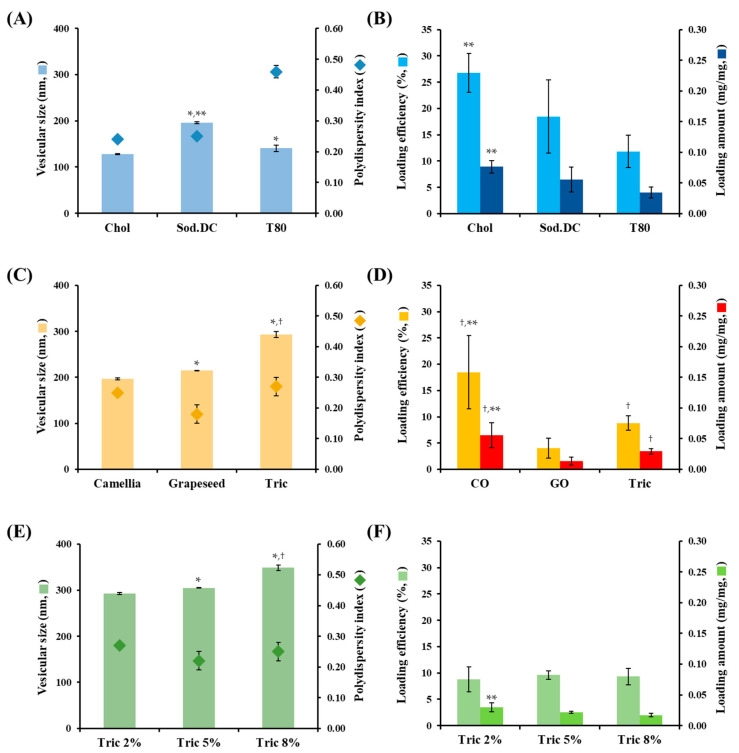
Effect of formulation variables on liposome vesicular size, polydispersity index, loading efficiency, and loading amount of Vit C. Effect of the type of edge activator on (**A**) vesicular size and polydispersity index and (**B**) Vit C loading efficiency and amount. Effect of kinds of neutral oil on (**C**) vesicular size, polydispersity index, and (**D**) Vit C loading efficiency and loading amount. Effect of the amount of Tric on (**E**) vesicular size and polydispersity index and (**F**) Vit C loading efficiency and loading amount. In (**A**,**B**), LOSs prepared with cholesterol (L1), sodium deoxycholate (L2), and polysorbate 80 were named Chol, Sod.DC, and T80, respectively. In (**C**,**D**), LOSs prepared with camellia oil (L2), grapeseed oil (L4), and tricaprylin (L5) were named CO, GO, and Tric, respectively. In (**E**,**F**), LOSs prepared with tricaprylin concentrations of 2% (L5), 5% (L6), and 8% (L7) were named Tric 2%, Tric 5%, and Tric 8%, respectively. The polydispersity index was obtained from the Zetasizer measurement. Data represent mean ± SD (n = 3). In (**A**,**B**), significant differences from Chol (** p* < 0.05) and T80 (*** p* < 0.05) were calculated using the ANOVA test. In (**C**,**D**), significant differences from CO (** p* < 0.05), GO (^†^
*p* < 0.05), and Tric (*** p* < 0.05) were calculated using the ANOVA test. In (**E**,**F**), significant differences from Tric 2% (** p* < 0.05), Tric 5% (^†^
*p* < 0.05) and T80 (*** p* < 0.05) were calculated using the ANOVA test.

**Figure 4 materials-16-02294-f004:**
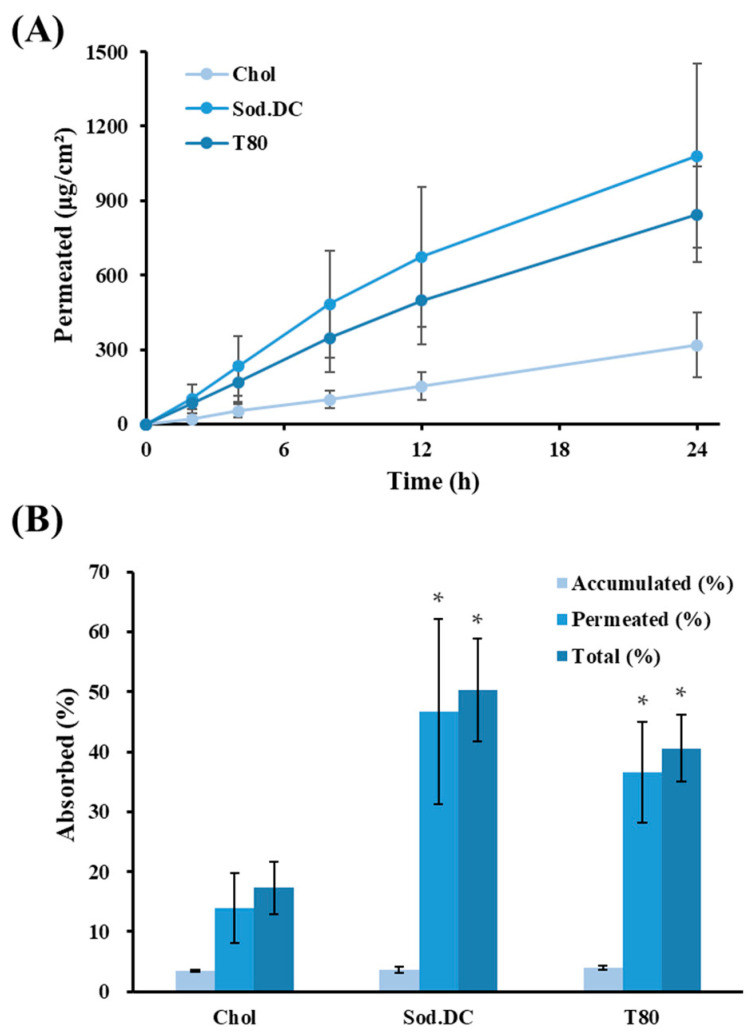
Ex vivo skin absorption of Vit C after topical administration of LOSs prepared with different edge activators. (**A**) Skin permeation profile (μg/cm^2^) of Vit C across porcine skin. (**B**) Percent of Vit C accumulated in the skin, permeated across the skin, and the sum of Vit C accumulated and permeated 24 h post-administration of LOSs. LOSs prepared with cholesterol (L1), sodium deoxycholate (L2), and polysorbate 80 (L3) were named Chol, Sod.DC, and T80, respectively. Total absorbed (%) is calculated by dividing the sum of the accumulated and permeated amounts of Vit C by the initially applied amount of Vit C. Data represent mean ± SD (n = 3). * represents significant difference from Chol (* *p* < 0.05).

**Figure 5 materials-16-02294-f005:**
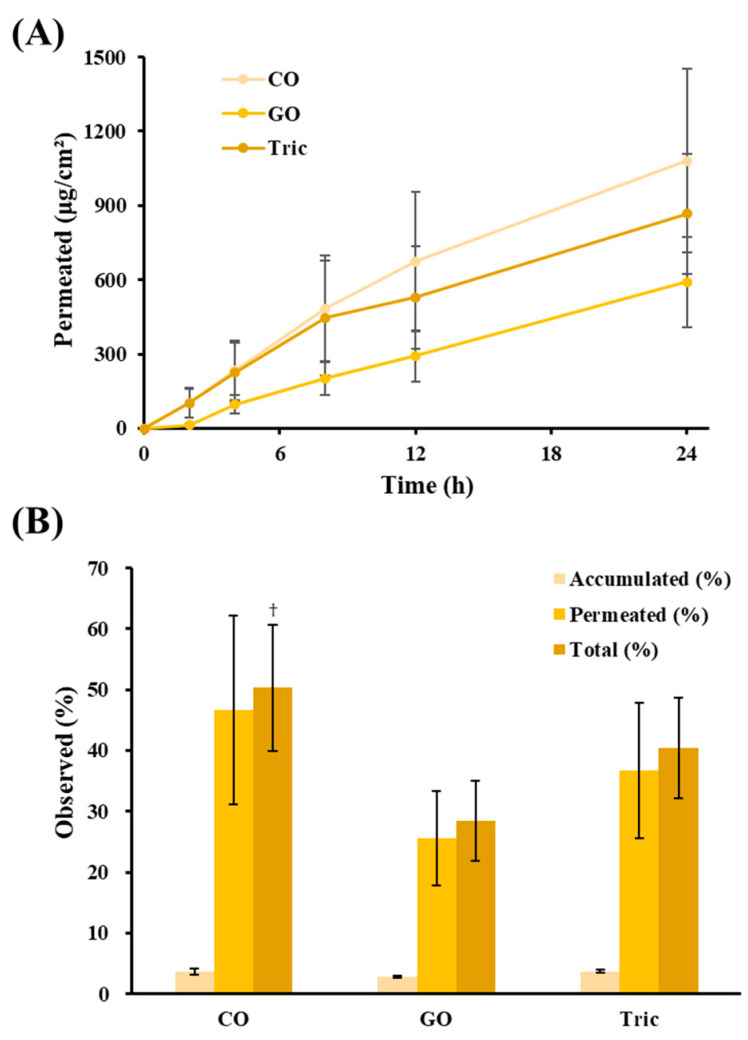
Ex vivo skin absorption of Vit C after topical administration of LOSs prepared with different oils. (**A**) Skin permeation profile (μg/cm^2^) of Vit C across porcine skin. (**B**) Percent of Vit C accumulated in the skin, permeated across the skin, and the sum of Vit C accumulated and permeated 24 h post-administration of LOSs. LOSs prepared with camellia oil (L2), grapeseed oil (L4), and tricaprylin (L5) were named CO, GO, and Tric, respectively. Total absorbed (%) is calculated by dividing the sum of the accumulated and permeated amounts of Vit C by the initially applied amount of Vit C. Data represent mean ± SD (n = 3). ^†^ represents significant difference from GO (^†^
*p* < 0.05).

**Figure 6 materials-16-02294-f006:**
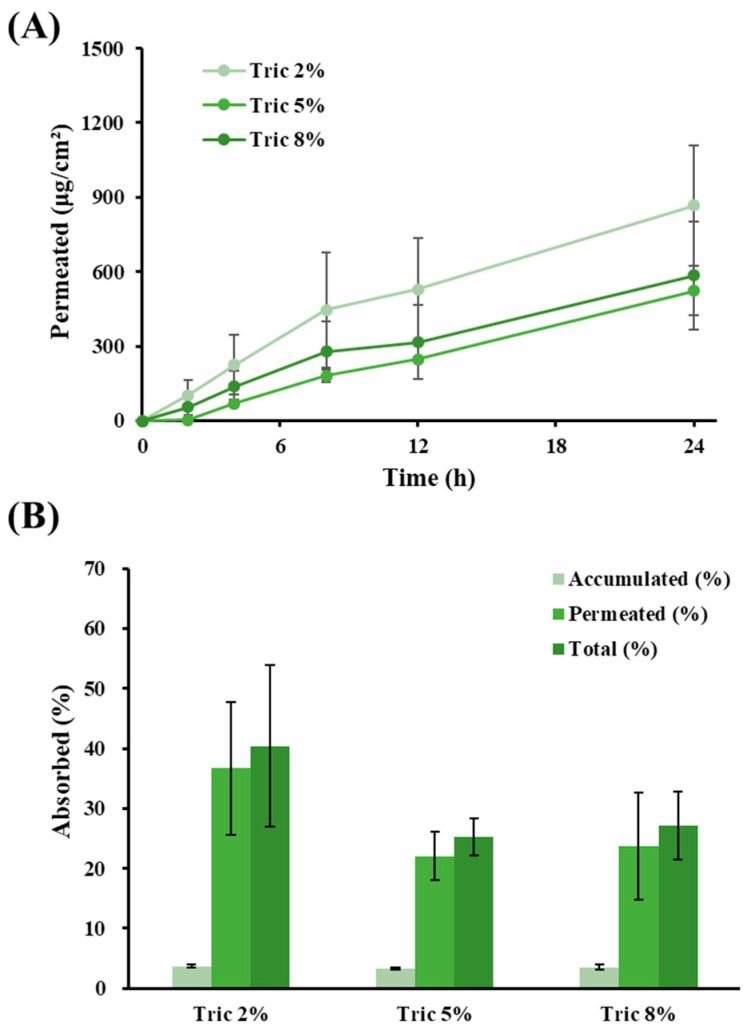
Ex vivo skin absorption of Vit C after topical administration of LOSs prepared with different amounts of oil. (**A**) Skin permeation profile (μg/cm^2^) of Vit C across porcine skin. (**B**) Percent of Vit C accumulated in the skin, permeated across the skin, and the sum of Vit C accumulated and permeated 24 h post-administration of LOSs. LOSs prepared with a tricaprylin concentration of 2% (L5), 5% (L6), and 8% (L7) were named Tric 2%, Tric 5%, and Tric 8%, respectively. Total absorbed (%) is calculated by dividing the accumulated amount, and the permeated amount of Vit C by the initially applied amount of Vit C. Data represent mean ± SD (n = 3).

**Table 1 materials-16-02294-t001:** Compositions of Vit C-loaded lipo-oil-somes (LOSs) formulated with different edge activators and neutral oils.

Formulation No.	L1	L2	L3	L4	L5	L6	L7
Vitamin C (mg)	200	200	200	200	200	200	200
Phosphatidylcholine (mg)	200	200	200	200	200	200	200
DPPG (mg)	4	4	4	4	4	4	4
Cholesterol (mg)	20	-	-	-	-	-	-
Sodium deoxycholate (mg)	-	20	-	20	20	20	20
Polysorbate 80 (mg)	-	-	20	-	-	-	-
Camellia oil (mg)	200	200	200	-	-	-	-
Grapeseed oil (mg)	-	-	-	200	-	-	-
Tricaprylin (mg)	-	-	-	-	200	500	800
10 mM succinate buffer	q.s.	q.s.	q.s.	q.s.	q.s.	q.s.	q.s.
Total (mL)	10	10	10	10	10	10	10
pH ^(a)^	3.5	3.2	3.4	3.2	3.3	3.2	3.2

^(a)^ pH data are expressed as mean (n = 3), and the SD values were smaller than 0.1 in all data.

**Table 2 materials-16-02294-t002:** Ex vivo permeation parameters of Vit C after topical administration of LOSs with different edge activators through porcine skins.

	Chol (L1)	Sod.DC (L2)	T80 (L3)
Flux (μg/cm^2^∙h)	13.0 ± 6.07	45.4 ± 15.3 *	38.7 ± 6.09 *
Lag time (h)	0.40 ± 0.07 ^†,^**	1.28 ± 0.63	1.32 ± 0.61
Permeability coefficient (10^−6^∙cm/h)	0.65 ± 0.21	2.27 ± 0.54 *	1.93 ± 0.23 *
Permeated (μg/cm^2^)	318.8 ± 131.8	1080.3 ± 370.3 *	845.2 ± 193.5 *

Notes: LOSs prepared with cholesterol (L1), sodium deoxycholate (L2), and polysorbate 80 (L3) are named Chol, Sod.DC, and T80, respectively. Data represent the mean ± SD (n = 3). *, ^†^, and ** represent significant differences from Chol (* *p* < 0.05), Sod.DC (^†^
*p* < 0.05), and T80 (** *p* < 0.05), respectively.

**Table 3 materials-16-02294-t003:** Ex vivo permeation parameters of Vit C after topical administration of LOSs with different neutral oils through porcine skins.

	CO (L2)	GO (L4)	Tric (L5)
Flux (μg/cm^2^∙h)	45.4 ± 15.3	25.3 ± 6.85	35.5 ± 9.29
Lag time (h)	1.28 ± 0.63	1.08 ± 0.38	2.27 ± 1.10
Permeability coefficient (10^−6^∙cm/h)	2.27 ± 0.54 ^†^	1.26 ± 0.23	1.78 ± 0.34
Permeated (μg/cm^2^)	1080.3 ± 370.31	592.2 ± 182.6	868.0 ± 242.3

Notes: LOSs prepared with camellia oil (L2), grapeseed oil (L4), and tricaprylin (L5) are named CO, GO, and Tric, respectively. Data represent the mean ± SD (n = 3). ^†^ represents significant difference from GO (^†^
*p* < 0.05).

**Table 4 materials-16-02294-t004:** Ex vivo permeation parameters of Vit C after topical administration of LOSs with different amounts of tricaprylin on porcine skins.

	Tric 2% (L5)	Tric 5% (L6)	Tric 8% (L7)
Flux (μg/cm^2^∙h)	35.5 ± 9.29	22.6 ± 4.00	23.9 ± 10.3
Lag time (h)	2.27 ± 1.10	0.12 ± 0.01 *	3.09 ± 4.38
Permeability coefficient (10^−6^∙cm/h)	1.78 ± 0.34 ^†^	1.13 ± 0.14	1.20 ± 0.39
Permeated (μg/cm^2^)	868.0 ± 242.3 ^†^	524.3 ± 100.2	585.2 ± 218.3

Notes: LOSs prepared with tricaprylin concentrations of 2% (L5), 5% (L6), and 8% (L7) were named Tric 2%, Tric 5%, and Tric 8%, respectively. Data represent the mean ± SD (n = 3). * and ^†^ represent significant differences from Tric 2% (* *p* < 0.05) and Tric 5% (^†^
*p* < 0.05), respectively.

## Data Availability

Not applicable.

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
