# Peer review of "Neutral Oil-Incorporated Liposomal Nanocarrier for Increased Skin Delivery of Ascorbic Acid"

_materials, 2023, doi:10.3390/ma16062294_

Round 1

Reviewer 1 Report

In this study, the authors developed a neutral oil-incorporated liposomal system to improve skin absorption of ascorbic acid, and the effect of edge activator and neutral oil on skin absorption of Vit C was also evaluated.

In general, the authors validate their hypothesis throughout the experiments, however, some minor concerns should be address prior to acceptance.

1.     On Page 6 Line 241, there is no “E” in the figure.   

2.     Normally, researchers used poly dispersity index (PDI) to refer the dispersion of the nanoparticles. In figure 3, the authors use homogeneity instead, is there any difference between homogeneity and PDI?

3.     The cell toxicity of VC loaded liposomes should be characterized since the transdermal drug delivery is the main application.

4.    The drug loading content should also provide in addition to drug loading efficiency.

Reviewer 2 Report

The manuscript materials-2268827 presents interesting research about better delivery of ascorbic acid in the skin by incorporating it in neutral oil liposomal nanocarrier.
This method can be used in the future to incorporate other active substances with similar physical and/or chemical properties, to facilitate their absorption in the skin and not only.

Reviewer 3 Report

In this study entitled (Neutral oil-incorporated liposomal nanocarrier for increased skin delivery of ascorbic acid), the authors prepared and evaluate different formulation of VIT C liposomes.  The studied the effect of different edge activators, different type of neutral oil, and different concentrations of oils on the properties of the prepared liposomes.

The manuscript is interesting however, there are aspects that should be better explored and explained before acceptance for publication.

Comments

1-    The manuscript should be revised regarding grammar and typo errors.

2-    What if the used stain while preparing the TEM samples?

3-    What is the angle of measurement of particle size?

4-    In line 174, didi the unit of drug concentration is correct?

5-    When you studied the effect of different edge activators on the amount of drug permeated, you explain your results due to difference in HLB. Please, add the HLB value of SDC, tween 80, and cholesterol.

6-    In line 148, what (LC) refer to?   

7-    In line 295, the sentence started by flux value while you wrote the lag time values. Please correct.

8-    In part 3.5. , why authors studied the effect of different concentrations of Tricaprylin while according to the previous section, Camellia oil gave better permeation and both camellia oil and grapeseed oil gave better particle size?

Reviewer 4 Report

The research paper is concerning to the development of new transdermal carriers of bioactive components. The authors describe a neutral oil-incorporated liposomal system  designed to improve skin absorption of ascorbic acid (Vitamin C), which is of big dermatological effects  improving the skin condition, possessing anti-aging  effects . protecting the skin from reactive oxygen species promoting collagen biosynthesis, reducing melanin synthesis, and enhancing the skin immunity . The effects of edge activator and  neutral oil incorporated in the formulation were evaluated. As components of the systems, Sodium deoxycholate, polysorbate  and cholesterol were discussed also of the components of proposed formulations as edge activators, and camellia oil, tricaprylin and grapeseed oil as neutral oils. The optimized lipid nanocarrier can be a promising tool  to promote skin absorption of Vit C, improving its dermatological functions.

The paper was writen clearly by a good scientific English, it can be published almost in present form after some minor English corrections.

Round 2

Reviewer 3 Report

Authors addresed all required changes and the manuscript is suitable now for publications